# Bone Marrow Clonogenic Myeloid Progenitors from *NPM1*-Mutated AML Patients Do Not Harbor the *NPM1* Mutation: Implication for the Cell-Of-Origin of *NPM1+* AML

**DOI:** 10.3390/genes11010073

**Published:** 2020-01-09

**Authors:** Rafael Diaz de la Guardia, Laura González-Silva, Belén López-Millán, Juan José Rodríguez-Sevilla, Matteo L. Baroni, Clara Bueno, Eduardo Anguita, Susana Vives, Laura Palomo, Helene Lapillonne, Ignacio Varela, Pablo Menendez

**Affiliations:** 1Department of Biomedicine, Josep Carreras Leukemia Research Institute, School of Medicine, University of Barcelona, 08036 Barcelona, Spain; rdiaz@carrerasresearch.org (R.D.d.l.G.); blopez@carrerasresearch.org (B.L.-M.); jrodsevilla@gmail.com (J.J.R.-S.); mbaroni@carrerasresearch.org (M.L.B.); cbueno@carrerasresearch.org (C.B.); 2Instituto de Biomedicina y Biotecnología de Cantabria, Universidad de Cantabria-CSIC, 39011 Santander, Spain; laura.gonzalezsilva@unican.es (L.G.-S.); ignacio.varela@unican.es (I.V.); 3Hematology and hemotherapy Department, Hospital Clínico San Carlos, IMDL, IdISSC, Departamento de Medicina, Universidad Complutense de Madrid, 28040 Madrid, Spain; eduardo.anguita@salud.madrid.org; 4Hematology Department, ICO-Hospital Germans Trias i Pujol, 08916 Badalona, Spain; svives@iconcologia.net (S.V.); lpalomo@carrerasresearch.org (L.P.); 5Josep Carreras Leukemia Research Institute, Universitat Autònoma Barcelona, 08193 Barcelona, Spain; 6Sorbonne Université, INSERM, Centre de recherche Saint-Antoine CRSA, AP-HP, Hôspital Armand Trousseau, Haematology Laboratory, F-75012 Paris, France; helene.lapillonne@aphp.fr; 7Centro de Investigación Biomédica en Red de Cáncer (CIBER-ONC), ISCIII, 08036 Barcelona, Spain; 8Instituciò Catalana de Recerca i Estudis Avançats (ICREA), Barcelona 08010, Spain

**Keywords:** AML, NPM1 mutations, FLT3-ITD, clonogenic progenitors, colony forming units (CFU)

## Abstract

The cell-of-origin of *NPM1*- and *FLT3*-mutated acute myeloid leukemia (AML) is still a matter of debate. Here, we combined in vitro clonogenic assays with targeted sequencing to gain further insights into the cell-of-origin of NPM1 and FLT3-ITD-mutated AML in diagnostic bone marrow (BM) from nine NPM1+/FLT3-ITD (+/−) AMLs. We reasoned that individually plucked colony forming units (CFUs) are clonal and reflect the progeny of a single stem/progenitor cell. NPM1 and FLT3-ITD mutations seen in the diagnostic blasts were found in only 2/95 and 1/57 individually plucked CFUs, suggesting that BM clonogenic myeloid progenitors in *NPM1*-mutated and NPM1/FLT3-ITD-mutated AML patients do not harbor such molecular lesions. This supports previous studies on NPM1 mutations as secondary mutations in AML, likely acquired in an expanded pool of committed myeloid progenitors, perhaps CD34−, in line with the CD34^−/low^ phenotype of NPM1-mutated AMLs. This study has important implications on the cell-of-origin of NPM1+ AML, and reinforces that therapeutic targeting of either *NPM1* or *FLT3-ITD* mutations might only have a transient clinical benefit in debulking the leukemia, but is unlikely to be curative since will not target the AML-initiating/preleukemic cells. The absence of NPM1 and FLT3-ITD mutations in normal clonogenic myeloid progenitors is in line with their absence in clonal hematopoiesis of indeterminate potential.

## 1. Cell-Of-Origin of NPM1-Mutated and FLT3-ITD-Mutated AML

Acute myeloid leukemia (AML) represents a heterogeneous group of malignant hematological disorders characterized by the rapid expansion of immature myeloid cells (blasts) in the bone marrow (BM). There is a wide disease heterogeneity in AML and patient risk-stratification principally relies on cytogenetic-molecular data [1]. AML frequently associates to chemotherapy refractoriness and relapse, suggesting failure of current therapies to eradicate leukemic initiating cells as a major mechanism underlying AML progression/relapse. AML is typically diagnosed without observation of a preleukemic phase so the cell-of-origin and the order of mutations remain poorly understood, particularly in normal karyotype AMLs (NK-AMLs) which represent 50%–60% of AML cases and lack a cytogenetic tag for single cell tracing. NK-AMLs are risk-stratified based on molecular biomarkers, such as NPM1 and FLT3-ITD mutations, which are found in 70% of NK-AML patients [2,3]. NPM1-mutations are well-documented in de novo AML, therapy-related AML, and also in donor cell-derived AML, supporting that NPM1 mutations are founder genetic alterations defining an independent AML entity [4,5].

Virtually all cancers are clonal and reflect the progeny of a single cell [6], but the evolutionary trajectory that leads from the initial somatic mutation to the eventual clinically overt cancer is not well mapped and suggest a complex/branching clonal architecture in many cancers. Thanks to the modern era of cancer genomics NPM1 and FLT3-ITD have now become ideal molecular tags for tracking lineage involvement and cell-of-origin in AML as they are usually very stably expressed in all leukemic cells [2,5,7].

The cell-of-origin of *NPM1*-mutated AML has long been a matter of debate. NPM1-mutated AMLs commonly lack expression of both CD34 [3,5]. The CD34^neg^ phenotype of NPM1-mutated AML raises questions as to whether the NPM1 mutation occurs in a CD34− committed myeloid progenitor or whether a rare pool of CD34+CD38− NPM1-mutated early progenitor/HSC exists. Falini’s group showed that MACS-sorted bulk CD34+ cells, and immature CD34+CD38− cells were NPM1-mutated by PCR and Western blot [8]. Of note, NPM1-mutated CD34+ cells recapitulated the AML phenotype when transplanted in bulk immunodeficient mice. In sharp contrast, another study from the same group revealed that despite the frequent involvement of two or more myeloid lineages, B-cell and T-cell lineages are not targeted by NPM1 mutations, indicating that NPM1 mutations may arise in a committed myeloid progenitor rather than in early/immature hematopoietic stem cells (HSC)/progenitors [9]. Later on, John Dick’s laboratory used high-coverage targeted-sequencing to demonstrate that highly purified HSCs and progenitors from NPM1-mutated AML patients do not harbor coincident NPM1 mutations present in AML blasts, proposing a model in which NPM1 mutations are acquired in an expanded pool of committed myeloid progenitors [10]. Additionally, Mel Greaves’s laboratory recently conducted a single cell analysis of the clonal architecture in AML and concluded that *NPM1* mutations are secondary to other AML driver mutations, and that the CD34+ cell fraction contains preleukemic subclones lacking NPM1 mutations [11].

Here, we combined in vitro clonogenic assays with targeted sequencing of both NPM1 and FLT3-ITD to gain further insights into the cell-of-origin of NPM1-mutated and FLT3-ITD-mutated AML in diagnostic BM from nine AMLs, five NPM1+/FLT3-ITD+, and four NPM1+/FLT3-ITD (Table 1). We reasoned that individually plucked colony forming units (CFUs) are clonal and reflect the progeny of a single cell HSC/myeloid progenitor. A low expression level of CD34 was confirmed in all NPM1-mutated AML patients (Table 1). An average of 55 ± 47 (range: 2–134) myeloid CFUs were obtained per AML patient, and morphological/phenotypic CFU scoring revealed multilineage representation of myeloid progenitors including immature mix-CFU (33% of the total), granulocyte colony-forming unit (G-CFU) (62%), and granulo-monocytic colony-forming unit (GM-CFU) (5%) (Table 1 and Figure 1A). We then confirmed that the CFU potential of BM cells from NPM1-mutated AML is exceptionally confined to the CD34-enriched population, and showed no correlation between the number of CFU and either percentage of blasts or percentage of CD34+ cells (Figure 1B,C). The multilineage clonogenic capacity and the lack of correlation between % of blasts/CD34+ cells indicate that CFU assays with BM-derived AML cells read out normal or preleukemic HSC/progenitors rather than self-renewing clonal AML blasts. Of note, the NPM1 and FLT3-ITD mutations seen in diagnostic blasts were found in only 2/95 (2%) and 1/57 (1.5%) individually plucked CFUs, suggesting that BM clonogenic myeloid progenitors from *NPM1*-mutated and NPM1/FLT3-ITD-mutated AML patients do not harbor either the *NPM1* or the FLT3-ITD mutation present in the AML blasts (Figure 1D). This study was IRB-approved (ref. HCB/2014/0687) by the Clinic Hospital of Barcelona and samples were accessed upon signed informed consent.

## 2. Conclusions

This study has implications on the cell-of-origin of NPM1+ AML. First, our data provide relevant information about the mutational status of NPM1 and FLT3-ITD in a clonal progeny derived from single HSC/myeloid progenitors. Second, our data supports previous studies suggesting that NPM1 mutations are secondary to other AML driver mutations acquired in an expanded pool of committed myeloid progenitors, perhaps CD34−, further supporting the CD34^−/low^ phenotype of NPM1-mutated AML patients. Third, despite leukemia-induced hematopoietic displacement, clonogenic/CFU assays from diagnostic BM from AML patients read out normal myeloid progenitors with multilineage representation rather than leukemic CFUs. Of clinical relevance, therapeutic targeting of either *NPM1* or *FLT3-ITD* mutations might thus have a transient benefit in restraining disease progression and debulking the leukemia but is unlikely to be curative since will not target the AML-initiating/preleukemic cells. Finally, the absence of NPM1 and FLT3-ITD mutations in normal clonogenic myeloid progenitors is in line with the fact that these particular mutations are rarely found in clonal hematopoiesis of indeterminate potential.

## Figures and Tables

**Figure 1 genes-11-00073-f001:**
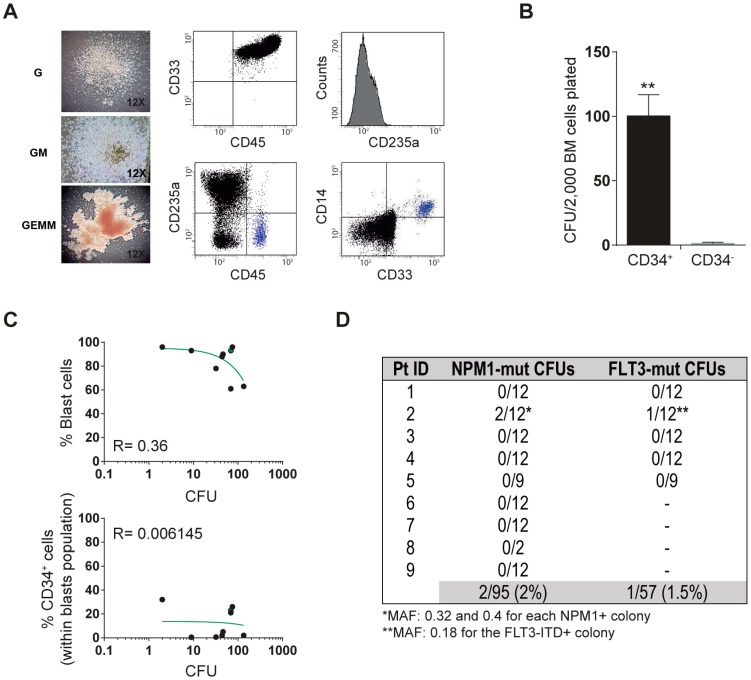
Characterization of clonogenic myeloid progenitors from bone marrow (BM) of NPM1-mutated acute myeloid leukemia (AML) patients. (**A**) Left, representative morphology of the indicated colony forming units (CFUs). HSC005 methylcellulose from VITRO SA (Madrid, Spain) was used. Right, representative immunophenotype of the indicated CFUs. (**B**) The CFU potential of BM cells from NPM1-mutated AML is exclusively confined to the CD34-enriched population. 2000 FACS-sorted CD34+ or CD34− cells were plated in duplicate. (**C**) Correlation between the number of CFU and both the % of blasts (top panel) or the % of CD34+ cells (bottom panel). 50,000 total BM mononuclear cells were plated in triplicate. (**D**) Summary of the targeted-sequencing of NPM1 and FLT3 in individually plucked CFUs from BM of n = 9 *NPM1*-mutated AML patients.

**Table 1 genes-11-00073-t001:** Clonogenic capacity of BM cells from *NPM1*-mutated AML patients.

Pt ID	Age	% Blasts	% Blasts CD34+	NPM1 Status	FLT3 Status	Total CFU *	CFU Mix	CFU G	CFU GM
1	52	93	23	mut	mut	69	19	50	0
2	48	90	5	mut	mut	46	4	42	0
3	63	61	21	mut	mut	69	40	29	16
4	41	96	26	mut	mut	75	6	69	0
5	42	93	0.5	mut	mut	9	2	7	0
6	18	88	2	mut	germline	44	27	17	0
7	62	63	2	mut	germline	134	47	71	16
8	39	96	30	mut	germline	2	0	2	0
9	67	78	0.6	mut	germline	32	13	19	0
Mean	48 ± 14	84 ± 12	12 ± 11			54 ± 47	33%	62%	5%

* Number of CFUs per 50,000 BM cells plated.

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
