# Peer review of "Bone Marrow Clonogenic Myeloid Progenitors from *NPM1*-Mutated AML Patients Do Not Harbor the *NPM1* Mutation: Implication for the Cell-Of-Origin of *NPM1+* AML"

_genes, 2020, doi:10.3390/genes11010073_

Round 1
Reviewer 1 Report
The paper is well written. Scientific evidence is well presented and briefly, but extensively discussed. Even if the number of the patients is small, this report is the first oservation, to the best of my knoweldge suggesting, that BM clonogenic myeloid progenitors in NPM1-mutated and NPM1/FLT3-ITD-mutated AML patients do not harbor such molecular lesions.
As this observation may represent an important stimulus for futher researches, both biological and clinical, I suggest to accept the manuscript in the present form.
Author Response
The paper is well written. Scientific evidence is well presented and briefly, but extensively discussed. Even if the number of the patients is small, this report is the first observation, to the best of my knowledge suggesting, that BM clonogenic myeloid progenitors in NPM1-mutated and NPM1/FLT3-ITD-mutated AML patients do not harbor such molecular lesions.As this observation may represent an important stimulus for further researches, both biological and clinical, I suggest to accept the manuscript in the present form.
We thank the referee very much for his/her compliments.

Reviewer 2 Report
Rafael et al. verified the gene mutations from single colonies derived from patient HSPC cells, proved John Dick’s conclusions from in vitro way. Overall this is not new but the convincing data may benefit our readership in such an area. There some minor issues needed to be addressed before considering publishing in our journal, here are my points:
The authors showed three different types of colonies in Table1, Mix, G, and GM. Please show the typical colony for each type in the figure. In Figure 1A, down panel, the colony has a high expression of CD235a but low CD45, it seems that this colony is an erythroid colony. Did authors add TPO in medium? How did authors define this colony? This colony seems not belonging to Mix, G, and GM.Author Response
Referee #2
The authors showed three different types of colonies in Table1, Mix, G, and GM. Please show the typical colony for each type in the figure.
We apologize for describing three different types of CFUs in Table 1 while showing only two types in Figure 1A. We have now added a representative CFU colony of each type: G, GM and Mix.
In Figure 1A, down panel, the colony has a high expression of CD235a but low CD45, it seems that this colony is an erythroid colony. Did authors add TPO in medium? How did authors define this colony? This colony seems not belonging to Mix, G, and GM
This colony is a mixed (GEMM) colony. It is composed by centered and well-defined “3D” groups of erythroid cells over a layer of M- and G- cells/clusters. G- clusters are composed of smaller cells in dense clusters while M-clusters are composed by bigger and more spread out cells. As it can be observed in the FACS plots, when plucked these colonies are made of CD235+CD45- (erythroid lineage cells) and CD45+CD14+ (myeloid cells).
We used commercial standard TPO-free methylcellulose medium (details are now provided in the main text, Lines 100 and 101). We copy below for the reviewer’s perusal representative images of multiple CFUs from the manufacturer’s. As observed, our representative colony looks like the CFU-Mix (CFU-GEMM).

This manuscript is a resubmission of an earlier submission. The following is a list of the peer review reports and author responses from that submission.